# COVID-19 Lockdown in Israel: The Environmental Effect on Ultrafine Particle Content in the Airway

**DOI:** 10.3390/ijerph19095507

**Published:** 2022-05-01

**Authors:** Einat Fireman Klein, Yotam Elimeleh, Yochai Adir, Lana Majdoub, Michal Shteinberg, Aharon Kessel

**Affiliations:** 1Pulmonary Division, Carmel Medical Center, Haifa 3436212, Israel; yochaiad@clalit.org.il (Y.A.); lana.m@campus.technion.ac.il (L.M.); michalsh@technion.ac.il (M.S.); 2Department of Internal Medicine B, Bnai Zion Medical Center, Haifa 3339419, Israel; yotamelime@gmail.com; 3The Technion-Institute of Technology, Rappaport Faculty of Medicine, Haifa 3525422, Israel; aharon.kessel@b-zion.org.il; 4Division of Allergy and Clinical Immunology, Bnai Zion Medical Center, Haifa 3339419, Israel

**Keywords:** EBC, UFP, lockdown, air pollution, NO_2_, COVID-19

## Abstract

Inhaled ultrafine particle (UFP) content in exhaled breath condensate (EBC) was observed as an airway inflammatory marker and an indicator of exposure to particulate matter (PM). The exceptional decline in air pollution during the COVID-19 lockdown was an opportunity to evaluate the effect of environmental changes on UFP airway content. We collected EBC samples from 30 healthy subjects during the first lockdown due to COVID-19 in Israel (March–April 2020) and compared them to EBC samples retrieved during April–June 2016 from 25 other healthy subjects (controls) living in the same northern Israeli district. All participants underwent EBC collection and blood sampling. Ambient air pollutant levels were collected from the Israeli Ministry of Environmental Protection’s online database. Data were acquired from the monitoring station closest to each subject’s home address, and means were calculated for a duration of 1 month preceding EBC collection. UFP contents were measured in the EBC and blood samples by means of the NanoSight LM20 system. There was a dramatic reduction in NO, NO_2_, SO_2_, and O_3_ levels during lockdown compared to a similar period in 2016 (by 61%, 26%, 50%, and 45%, respectively). The specific NO_2_ levels were 8.3 ppb for the lockdown group and 11.2 ppb for the controls (*p* = 0.01). The lockdown group had higher UFP concentrations in EBC and lower UFP concentrations in serum compared to controls (0.58 × 10^8^/mL and 4.3 × 10^8^/mL vs. 0.43 × 10^8^/mL and 6.7 × 10^8^/mL, *p* = 0.05 and *p* = 0.03, respectively). In this observational study, reduced levels of air pollution during the COVID-19 lockdown were reflected in increased levels of UFP airway contents. The suggested mechanism is that low airway inflammation levels during lockdown resulted in a decreased UFP translocation to serum. Further studies are needed to confirm this hypothesis.

## 1. Introduction

Ambient air pollution is a leading cause of morbidity and mortality, accounting for about 4.2 million premature deaths annually worldwide [1]. Traffic-related air pollution (TRAP) is a major source of pollution in urban areas, leading to a wide range of adverse health effects. It consists of a mixture of vehicular exhausts and evaporative emissions, as well as secondary pollutants produced in the atmosphere. NO_2_ is an important surrogate of TRAP since it enters the atmosphere primarily from the burning of fuel [2]. As opposed to atmospheric ozone, ground-level ozone is formed when sunlight reacts with pollutants, such as NO_x_, from vehicular emissions [1]. Particulate matter (PM) is a widespread air pollutant, consisting of a mixture of airborne particles varying in size, chemical properties, and composition [3]. Ultrafine particles (UFP) are toxic due to their small size and penetration into deeper lung compartments [4]. The association between chronic exposure to PM and an increased risk of developing cardiovascular and respiratory diseases is supported by large-scale epidemiological studies [5,6].

Environmental monitoring of air pollution does not take into consideration the individual’s levels of daytime exposure and the effect on health status. In addition, most monitoring stations do not provide comprehensive data on UFP levels due to the challenges of monitoring and characterizing these atmospheric particles [7]. In contrast, biological monitoring detects the extent of individual exposure to environmental pollutants by measuring the levels of the selected substances and the breakdown of their components in tissues and fluids. Many components of the human exposome can be measured in various biological samples, such as blood, sputum, and exhaled breath condensate (EBC). EBC is a simple non-invasive tool, which can be used to reflect airway epithelial function. The EBC is a matrix that consists of volatile and non-volatile components that originate mainly from airway lining fluid (ALF). EBC consists of 99.9% condensed vapor and 0.1% exhaled droplets entrapping nonvolatile compounds at extremely low concentrations [8]. The evaluation of airway inflammatory diseases by means of EBC has been studied extensively. Elevated H_2_O_2_ and lower PH levels were found in the EBC of COPD and asthma patients compared to healthy subjects [9], and further pH reduction was identified during COPD and asthma exacerbation [10,11]. There are, however, no normal reference values for EBC biomarkers. Current knowledge is derived from small studies with sometimes contradictory results, likely due to the lack of reproducibility of EBC biomarkers [8].

Our recent study [12] was designed to evaluate UFP as a marker of airway inflammation. EBC and serum were collected from COPD patients and healthy smoker and non-smoker controls. Particle sizes and concentrations in biological samples were analyzed and correlated to clinical parameters and oxidative stress markers in EBC. COPD patients had lower UFP concentrations in EBC and higher ones in serum compared with controls. In addition, EBC UFP concentrations were negatively correlated with pack-years and positively correlated with FEV1 and DLCO. Low EBC UFP concentrations and CRP levels ≥5 mg/L were independent predictors of the frequent exacerbator phenotype. Our hypothesis is that an increased epithelial permeability in inflamed airways will allow more UFP to penetrate and translocate into the lung parenchyma and circulation, leading to a reduced concentration of UFP in EBC samples [12]. We also revealed that low UFP concentrations in EBC served as an indicator of high-level PM exposure [13].

The novel coronavirus (COVID-19) was first reported in Wuhan, China in December 2019 and then spread rapidly worldwide [14]. Many countries placed drastic restrictions upon public behavior several times throughout 2020, ranging from social distancing and reduction of large gatherings to the point of complete closure of workplaces and lockdowns with stringent exceptions. As a result, non-essential factories were shut down, and flights and other means of transportation were restricted, leading to the improvement of air quality in many of the world’s cities and reducing environmental pollution [15]. Studies spanning multiple continents have shown decreases of 50% in NO_2_ concentrations, 40% in PM_2.5_ concentrations, and 60% in PM_10_ concentrations during global lockdowns [16,17]. This dramatic decline in air pollution during COVID-19 lockdown periods provided an exceptional opportunity to observe the effect of environmental changes on UFP airway content and to evaluate it as a potential indicator of the level of exposure to air pollutants.

## 2. Materials and Methods

### 2.1. Study Population

Thirty healthy volunteers recruited from healthcare staff members at Bnai Zion Medical Center (Haifa, Israel) were enrolled 4 weeks after the first Israeli national COVID-19 lockdown was installed in April 2020, and they comprised the “lockdown” group. The control group consisted of 25 volunteers from the same medical center who had been recruited for another study from April–June 2016. The original study population included 40 healthy subjects. Only 25 individuals under the age of 50 years were included in the study in order to create similar-aged controls [12]. All the participants from both groups resided in the same district in the northern part of Israel. Individuals with chronic lung diseases or significant smokers (>15 PY) were excluded. We used a demographic, occupational, and health data self-reported questionnaire. All study participants underwent EBC testing and blood sampling. Participants were asked to avoid smoking and the use of any type of inhaled medication prior to EBC collection, as recommended by EBC guidelines [18]. The study was approved by the Helsinki committee of the Bnai Zion Medical Center (nos. BNZ-0051-14 and BNZ-0163-20), and all subjects signed their informed consent to participate.

### 2.2. EBC Sample Collection

We used a portable condenser (a transportable unit for research on biomarkers obtained from disposable exhaled condensate collection systems; TURBODECCS; ItalChill, Parma, Italy), which is specifically designed to collect EBC in clinical and workplace settings. The use of TURBO-DECCS was validated in this context [18]. Briefly, subjects breathed into the collecting system for 10–15 min at normal tidal volume. Samples were stored at −80 °C until analysis. All EBC collections were performed in an environment with room temperature (22–23 °C) and humidity (50%) controlled by a closed air-conditioning system.

### 2.3. Peripheral Blood Samples

Blood samples were drawn by conventional methods. The samples were analyzed for high-sensitivity C-reactive protein (CRP) in the chemistry labs at the Bnai Zion Medical Center.

### 2.4. UFP Measurement and Analysis in Biological Samples

The size and concentration of the particles in EBC and serum samples were assessed with the NanoSight LM20 system (NanoSight Ltd., Salisbury, UK) by a method that processes and estimates the size and the concentration of particles in liquid solution based on the Brownian motion and the speed of the particles: the smaller the particles, the faster they move and the further the distance that they travel. The particles contained in the samples were in the range of 10–1000 nm, and they were visualized by virtue of the light they scattered when illuminated by a laser light source. The Nanoparticle Tracking Analysis (NTA software version 3.4, NanoSight Ltd., Salisbury, UK) tracked many particles individually and calculated their size according to their speed of motion. Approximately 0.3 mL of EBC/serum samples were introduced into the viewing unit which produced a readout of the output of total particle concentrations, the percent of particles in the nano-size range, and the mean particle sizes. All serum samples were diluted 1:10,000 by means of double-filtered distilled water. UFP analyses were performed in the lung and allergy laboratory at the Tel Aviv Medical Center (Tel Aviv, Israel).

Characterization of exhaled particles from the healthy subjects revealed that during normal tidal breathing (such as that for EBC sampling), reproducibility within subjects was high, with a count median diameter of 278 nm [19]. Similarly, the particle size distribution was 256 ± 81 nm in the control group vs. 223 ± 78 nm in the lockdown group (*p* = 0.15). This high reproducibility indicates that the individual particle number is highly characteristic of the actual individual lung status. Additionally, using an EBC device with a saliva trap contributed to greater heterogeneity in particle size by removing particles from the oral cavity.

### 2.5. Environmental Ambient Air Pollution Data

Air pollutant levels were collected from the Israeli Ministry of Environmental Protection’s online database https://www.svivaaqm.net/ (accessed on 1 July 2020).

The data were collected from 17 general and transportation monitoring stations located in a land area of 1000 km [2] in the northern part of Israel. The “general” monitoring stations are located at roof height in representative areas, none of which are adjacent to specific emission sources, such as industrial plants. The “transportation” monitoring stations are located at pavement height near primary traffic junctions. All monitoring stations are run by the Ministry of Environmental Protection and the Israel Electric Corporation. Concentrations of the air pollutants PM_2.5_, SO_2_, NO_X_, NO_2_, NO, and O_3_ were collected from all monitoring stations. The data retrieved from the monitoring station closest to each subject’s home address were calculated as an average of the results derived 1 and 3 months prior to EBC performance. This study focused on a 1-month period since enrollment of the subjects began 4 weeks after the first lockdown had been implemented The relevant monitoring station was located within a radius of up to 11 km from each subject’s residential area, which had been reported to represent the radius most sensitive to environmental changes [20]. The mean ± standard deviation (SD) distance from the monitoring station was 5 ± 0.65 km for the control group and 6.2 ± 1.3 km for the lockdown group (*p* = 0.4).

### 2.6. Statistical Analysis

Statistical analyses were performed with the SPSS software version 25.3 (for Windows (Issued to Tau.ac.il, Tel Aviv University, Israel). Differences between air pollutants and biological monitoring levels in the lockdown group were compared to controls by the *t*-test. Data analysis was performed by analysis of covariance (ANCOVA) with age adjustment for group mean differences. Values were considered significant at *p* ≤ 0.05. Using independent samples *t*-test in observational studies is possible assuming there are no time-varying variables that are systematically related to the outcome of interest [21]. Modern approaches to causal inference may help facilitate accurate inference in the presence of unmeasured and time-varying confounding variables [22]. However, a recent systemic review of infectious disease observational studies found that these modern causal methods were not being implemented. Interdisciplinary collaborations between statisticians and researchers are needed [23].

Normal distribution of variables was examined by the Shapiro–Wilk test. Spearman’s correlation was used since all continuous variables had abnormal distributions. The sample size was calculated based on previous study results [2]. The minimal sample size was calculated to detect a mean difference of 0.2 (ΔMe = 0.2) between UFP-EBC concentrations of the subjects in the lockdown and control groups to account for α = 5% and 80% power. The SD for the healthy group in a pilot study was 0.28. Based on these values, 17 subjects would need to be included in each group in order to detect a mean difference of 0.2 (ΔMe = 0.2) in UFP-EBC concentration between the groups. Since there were 25 healthy subjects who served as the control group, we decided to recruit at least 25 healthy subjects for the lockdown group.

## 3. Results

### 3.1. Study Population

The demographic and clinical characteristics of the study population are shown in Table 1. The 55 healthy subjects were divided into the lockdown group (*n* = 30), whose members were enrolled 4 weeks after the first Israeli national lockdown was installed, and the control group (*n* = 25), whose members were derived from participants of an earlier study [8,9], whose samples were retrieved during April–June 2016, and who lived in the same northern Israeli district. There were no differences between the groups other than the lockdown group being significantly younger than the control group (34 ± 6.3 years compared to 41.9 ± 4.4 years, respectively, *p* = 0.001). All results were subsequently adjusted to age.

### 3.2. Air Pollutants

The data on air pollutant levels were collected and calculated from 17 monitoring stations as an average of 1-month pre-EBC analysis, including PM_2.5_, SO_2_, NO_X_, NO_2_, N_O_, and O_3_. The lockdown group was exposed to lower levels of TRAP, as shown in Table 2. There was a dramatic reduction in NO, NO_2_, SO_2_, and O_3_ levels during the lockdown in 2020 compared to the same period in 2016 (61%, 26%, 50%, and 45% reductions, respectively). Figure 1A,B show the reduction in NO_2_ exposure levels during the lockdown as measured in environmental stations located near the subjects’ home addresses. The PM_2.5_ exposure level of the lockdown group was lower than that of the control group only when calculating the 3-month average prior to EBC collection (15.1 µg/m^3^ vs. 16.3 µg/m^3^, *p* = 0.04).

### 3.3. Ultrafine Particles in EBC and Serum

EBC and serum samples were collected from each subject in both groups. The UFP concentration was measured in EBC and serum samples, and CRP levels were also measured in serum. Age, gender, and smoking status had no significant effect on UFP concentrations in the biological samples. Biological monitoring of EBC-UFP content in relation to environmental station location revealed a mirror image of the differences in the NO_2_ level between the two study groups (Figure 1). The lockdown group had higher UFP concentrations in EBC compared to the controls (0.58 × 10^8^/mL vs. 0.43 × 10^8^/mL, respectively, *p* = 0.05) (Figure 2A). Serum UFP concentrations of the lockdown group subjects were lower compared to the controls (4.3 × 10^8^/mL vs. 6.7 × 10^8^/mL, *p* = 0.03) (Figure 2B). Similarly, the serum CRP levels of the lockdown group were lower compared to the controls (0.9 mg/L vs. 3.1 mg/L, *p* = 0.056) (Figure 2C). The CRP level correlated positively with age in the study population (r = 0.4, *p* = 0.001). All of these results were adjusted to age. Correlation of the air pollutant data with the CRP levels and UFP concentrations in EBC and serum revealed that the exposure levels of TRAP (e.g., NO_2_, SO_2_, and O_3_) correlated positively with the CRP and UFP concentrations in serum of the entire study population (Table 3). There was a negative correlation between UFP concentrations in EBC and serum, but it was not statistically significant (r = −0.21, *p* = 0.14). No correlations were found between UFP-EBC concentrations and air pollutant levels.

## 4. Discussion

The sharp decline of air pollutant levels during the COVID-19 lockdown was reflected in UFP concentrations in the EBC of a healthy population. This descriptive study of a short-term exposure effect was carried out during a COVID-19 lockdown in Israel. This unusual period made it possible for us to observe the effects of significant environmental changes. These exceptional changes in air quality also gave us the opportunity to compare the data of this period with equivalent data that had been collected in the same regions during ordinary conditions of air pollution. There was a significant reduction in NO, NO_2_, SO_2_, and O_3_ levels during lockdown compared to the same period in 2016 (reductions of 61%, 26%, 50%, and 45%, respectively). PM_2.5_ exposure level in the lockdown group was lower than in the control group only when calculating the average of the 3 months prior to EBC, while the 1-month average showed no group differences.

The exposure level for both groups exceeded the annual PM_2.5_ mean exposure recommended by the WHO guidelines [1]. Consistent with our finding, a recent global observational analysis by Sokhi et al. demonstrated a positive correlation between the reductions in NO_2_ and NO_x_ concentrations and the mobility of people during lockdowns [24]. In a study conducted in several European cities, lockdown caused NO_2_ reductions of up to 58% in urban areas and up to 44% in rural areas. Similarly, a reduction of up to 70 % in NO_2_ levels was observed during a lockdown in India [25,26]. There was also a reduction of 25–60% in SO_2_ in both regions during 2020 compared to 2015–2019. No consistent changes in O_3_ patterns were observed in 2020, primarily due to different O_3_ formation regimens_._ The findings for PM_2.5_ were mixed, even within the same region, possibly resulting from secondary PM formation and long-range transport of dust or biomass burning [21].

Exhaled particle analysis was recently suggested as one of the techniques for the functional assessment of the small airways [27]. Our evaluation of small airway status during lockdown by measuring UFP content in EBC and serum of healthy subjects showed that the lockdown group had higher UFP concentrations in EBC and lower UFP concentrations in serum compared to controls. This is consistent with our previous observation that low UFP content in EBC could be an indicator of high levels of pollutant exposure [13]. We had also recently demonstrated that low UFP levels in EBC indicated high levels of airway inflammation, as evidenced by low UFP concentrations in EBC and high UFP concentrations in the serum of COPD patients compared to healthy controls [12]. Few previous studies have measured UFP concentration in EBC using Nanosight. Benor et al. found no differences in UFP-EBC concentrations between asthmatic children and non-asthmatic children [28]. The suggested mechanism behind our observation is the increased permeability of inflamed respiratory epithelium (Figure 3). UFP-induced oxidative stress can lead to an increase in permeability of the lung epithelium, which then allows particles and particle-loaded macrophages to penetrate the interstitium [29]. Accordingly, an in vitro study showed that a decrease in the tight junctional resistance of mouse alveolar epithelial cell monolayers caused a drastic increase in the translocation of engineered nanoparticles across the epithelial barrier [30]. Several studies, both in human and animal models, have shown that inhaled nanoparticles can rapidly translocate into the circulation and accumulate in extrapulmonary organs [31,32,33,34]. The concept that UFP translocate to the circulation was also proposed by the findings of a murine model in which the induction of lung inflammation resulted in a shift of the UFP pattern in bronchoalveolar lavage towards larger particles, a process which can be explained by translocation of smaller particles thorough inflamed epithelium [35]. This was further demonstrated in a study analyzing UFP content in biological samples retrieved from artificial stone dust workers, a positive correlation was found between UFP serum concentrations and the severity of silicosis CT findings [36]. This was reflected in the current study, with air pollutant levels being positively correlated with both serum UFP concentrations and serum CRP levels.

The association between long-term exposure to air pollution and increased risk of COVID-19 infection and disease severity was reviewed in a recent joint workshop report of ERS, ISEE, HEI, and WHO [37]. The direct cellular damage and increased lung inflammation induced by PM exposure were proposed as comprising the mechanism behind this association [37]. Such an association may now be further supported by our current findings of increased UFP-EBC levels and decreased UFP-serum levels during periods of low air pollution exposure.

This study has several limitations that bear mention. It is a proof of concept and not a national-level study due to the small sample size. In addition, the UFP content in EBC was not monitored within the same study group, but rather between two different study groups. This was due to the fact that most of the members of the first 2016 control group were no longer available during the lockdown period since most of them had moved to various parts of the country. Large-scale and prospective studies evaluating UFP content in EBC in the same individual after exposure to various levels of air pollution are warranted. Moreover, unlike the control group, the lockdown group could not undergo lung function tests because of strict restrictions set down at the onset of the COVID-19 pandemic. Spirometry was considered to be an aerosol-generating procedure [38], and therefore prohibited during this period. However, both the control group and the study group included the same type of population of normal healthy healthcare volunteers with the same habits and demographic characteristics.

## 5. Conclusions

Reduced levels of air pollution during COVID-19 lockdowns were reflected in increased levels of UFP airway contents. High UFP concentrations in EBC and low concentrations in the serum of healthy subjects during lockdown support our hypothesis that increased epithelial permeability could be the mechanism behind our findings.

## Figures and Tables

**Figure 1 ijerph-19-05507-f001:**
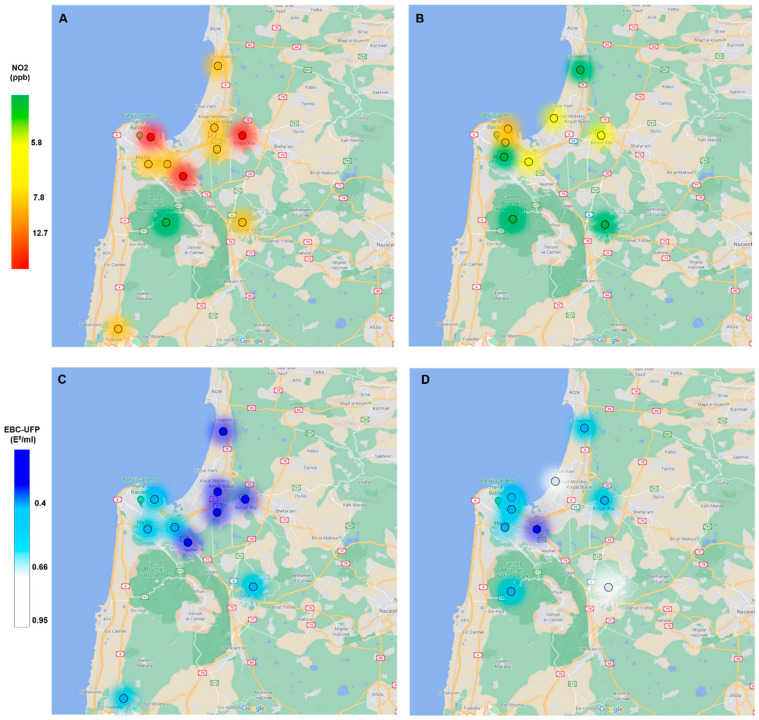
NO_2_ and EBC-UFP levels in the lockdown group vs. the control group. NO_2_ exposure level as measured in environmental stations located near the subjects’ home addresses during April 2016 (**A**) and during the first lockdown in April 2020 (**B**). NO_2_ 1-month average exposure level (ppb); green dots x ≤ 5.8, yellow dots 5.8 < x ≤ 7.8, orange dots 8.7 < x ≤ 12.7, red dots x > 12.7. Biological monitoring of EBC-UFP content in relation to environment stations during April 2016 (**C**) and during April 2020 (**D**). UFP concentrations in EBC (10^8^/mL); blue dots x ≤ 0.4, light blue dots 0.4 < x ≤ 0.66, white dots 0.66 < x ≤ 0.95.

**Figure 2 ijerph-19-05507-f002:**
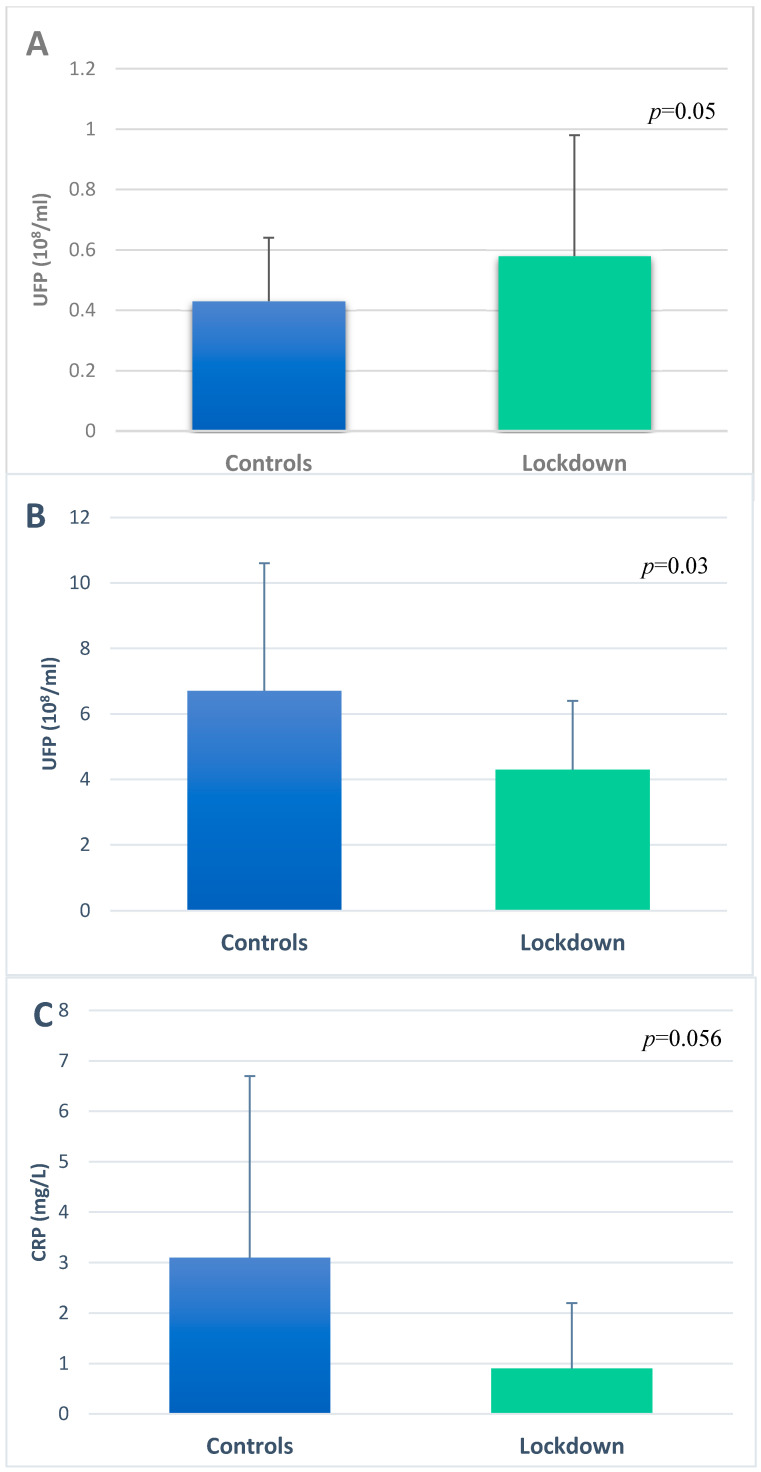
UFP and CRP levels in biological samples of the study population (*n* = 55). (**A**) Ultrafine particle concentrations in exhaled breath condensate and (**B**) in serum. (**C**) C-reactive protein concentrations in serum.

**Figure 3 ijerph-19-05507-f003:**
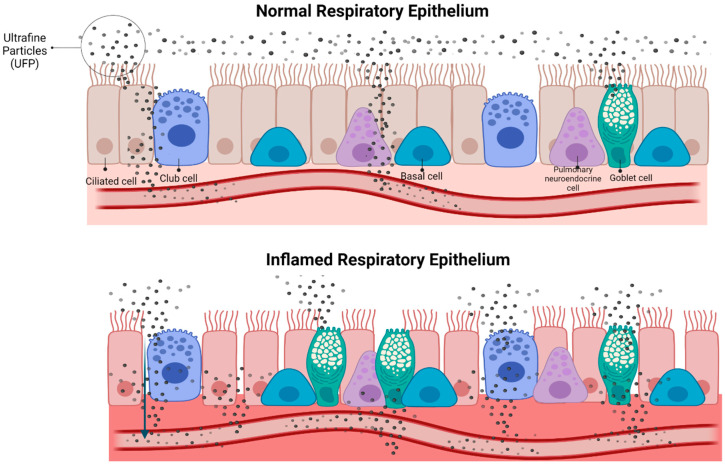
UFP content in inflamed respiratory epithelium-suggested mechanism. Increased epithelial permeability in inflamed airways allows more UFP to penetrate and translocate into the lung parenchyma and circulation, leading to a reduced concentration of UFP in EBC samples. Created with BioRender.com.

**Table 1 ijerph-19-05507-t001:** Demographic and clinical characteristics of the study population (*n* = 55).

	Control Group (*n* = 25)	Lockdown Group (*n* = 30)
* Age, years (mean, range)	41.9, 35–49	34, 22–48
Male, *n* (%)	14 (56)	10 (33)
Smokers ^a^, *n* (%)	12 (48)	12 (40)
Pack years, (mean ± SD)	3.8 ± 5.5	1.7 ± 3.5
Residence facing road, *n* (%)	11 (44)	19 (63)
Distance from monitoring station, km (mean ± SD)	5 ± 3.3	6.2 ± 7.1

* *p* < 0.05; ^a^ Active or former smokers with ≤15 PY history. SD-standard deviation.

**Table 2 ijerph-19-05507-t002:** TRAP exposure level of the study population (*n* = 55).

TRAP, ppb (Mean ± SD)	Control Group (*n* = 25)	Lockdown Group (*n* = 30)	*p*-Value ^+^
SO_2_	0.7± 0.64	0.36 ± 0.14	0.037
NOX	13.65 ± 8.7	6.7± 2.9	0.01
NO_2_	11.24± 5	8.3± 3.2	0.01
NO	3.25 ± 3.6	1.27 ± 1.1	0.014
O_3_	62.4 ± 17.4	34.1 ± 2.2	<0.001

TRAP (traffic-driven air pollutants) levels were calculated from monitoring stations close to each subject’s residence and presented as the mean of measurements performed 30 days prior to study recruitment. PPB—particles per billion; SD—standard deviation; SO_2_—sulfur dioxide; NO_X_—nitrogen oxide; NO_2_—nitrogen dioxide; NO—nitrogen oxide; O_3_—ozone. ^+^
*p* < 0.05 was considered significant using.

**Table 3 ijerph-19-05507-t003:** Serum UFP and CRP levels of study population (*n* = 55) in correlation with TRAP.

	Serum UFP Concentration (10^8^/mL)	Serum CRP Level (mg/L)
NO_2_—1 month of exposure (ppb)	r = 0.3, *p* = 0.04	r = 0.23, *p* = 0.09
SO_2_—1 month of exposure (ppb)	r = 0.2, *p* = 0.27	r = 0.26, *p* = 0.14
O_3_—1 month of exposure (ppb)	r = 0.34, *p* = 0.06	r = 0.64, *p* < 0.001

TRAP (traffic-driven air pollutants) levels 30 days prior to study recruitment were calculated from monitoring stations close to each subject’s residence. UFP—ultrafine particles; SO_2_—sulfur dioxide; NO_X_—nitrogen oxide; NO_2_—nitrogen dioxide; NO—nitrogen oxide; O_3_—Ozon; CRP—C-reactive protein. *p* < 0.05 was considered significant.

## Data Availability

The data presented in this study are available on request from the corresponding author. The data are not publicly available due to ethical reasons.

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
