# Peer review of "COVID-19 Lockdown in Israel: The Environmental Effect on Ultrafine Particle Content in the Airway"

_ijerph, 2022, doi:10.3390/ijerph19095507_

Round 1
Reviewer 1 Report
The manuscript is definitely improved and many mistakes have been corrected. The raw results are needed to be proved given so many mistakes in the initial submitted manuscript. Further, as there were so many corrections in the revised manuscript, I think it is necessary to justify the raw results. Also, regarding figure 2, which shows the UFP and CRP levels in biological samples of the study population, statistic doesn’t look as good as the numbers shown, therefore verification of the raw results is very necessary.
Author Response
We appreciate your valuable critique and the observation of several elements that needed clarification. The raw data and SPSS files for each table and figure are attached as requested. In addtion we made every effort to improve the english.
Please see the attachment

Reviewer 2 Report
The revised manuscript made sufficient changes based on my comments. The authors have provided clarification in the methods and results section. I suggest to publish this manuscript.
A minor issue is: please increase the resolution of Figure 3, and remove the web advertising note on the bottom right. Labels and introduction are also recommended to add (e.g. which color represent which type of the cell) to make it more clear.
Author Response
We thank the reviewer for pointing out another valuable comment. The figure resolution was improved and cell type labels were added as recommended.
Please see the attachment.

Reviewer 3 Report
Dear Authors,
thank you for having addressed my main comments. Now the aim of the paper is much more clear to me and is clearly stated. However, some doubts regarding the methods still remain. As already suggested in the previous report, I suggest you integrating a bit the existing literature on statistical methods applied to observational and causal studies in medicine and epidemiology. I don't want to suggest specific methodology, just extend the current paper comparing what you do with other approaches and explain why your methods are acceptable (given the scope).
Cheers!
Author Response
Our thanks for providing us with a valuable critique and for pointing out the need for further clarification. As recommended, we integrated updated literature on statistical methods used in observational and causal studies in medicine.
"Using Independent samples t-test in observational studies are possible assuming there are no time-varying variables that are systematically related to the outcome of interest. Modern approaches to causal inference may help facilitate accurate inference in the presence of unmeasured and time-varying confounding variables. However, a recent systemic review of infectious disease observational studies found that these modern causal methods were not being implemented. Interdisciplinary collaborations between statisticians and researchers are needed."
(Now appears on lines 174-180)
Please see the attachment

Round 2
Reviewer 1 Report
Accept in present form
This manuscript is a resubmission of an earlier submission. The following is a list of the peer review reports and author responses from that submission.
Round 1
Reviewer 1 Report
This is a report showing reduced levels of air pollution during COVID-19 lockdowns with increased levels of UFP airway contents and reduced concentrations in the serum of healthy subjects during lockdown. There were some very interesting and promising results. However, the report showed some contradictive results to the conclusion. More raw data would be helpful to verify the figures shown in the submitted paper. Anyway, the paper will benefit taking the following remarks into account.
1, Abstract could be improved. Please refine the abstract. Some of the information such as Nanosight supplier is not necessary. There are some typos in the last few sentences as well.
2, In the discussion, it is said there are several limitations. The current number of the study population is only 55, which should be improved for such a national-level study. If not, please explain. Please also add more contents such as the following work etc.
3, In Table2, the exposure level for NO, NO2, SO2, and O3 are all lower for Control group but not Lockdown group, whereas the description said a dramatic reduction during lockdown. Please explain.
4, There have been some studies from different areas of the world. Please compare the results to other similar study in terms of UFP contents from Nanosight etc as well as the levels in NO, NO2. SO2, and O3 etc.
5, In the Conclusion, it is said “increased epithelial permeability could be the mechanism behind our findings”. Please explain. If there is low concentration of UFP in the serum of healthy subjects during lockdown, should it indicate the epithelial permeability decreased?
Reviewer 2 Report
“COVID-19 lockdown in Israel: Environmental effect on ultrafine particles content in airway” by Klein et al. performed consecutive study to investigate the relationship among UFPs in EBC and serum, CRP in the serum, and novel ambient pollutants level. The study focused on an interesting topic and is worthwhile for publishing. I think the topic fulfill the scope of IJERPH journals.
I do have a couple major concerns and minor comments on it before publishing.
Major concern:
The human expired aerosol size distributions are largely variable (https://www.sciencedirect.com/science/article/abs/pii/S0021850211001200). How did the study limit the influence of variation of the exhale particle size from person to person? How is the EBC particle vary with study group/ control group?
Minor revision:
Introduction:
Line 57. The exhaled breath condensate (EBC) is a core term in this article. I suggest to discuss a little more about it. E.g. definition, core studies about EBC and human health relationship.
Line 62. Ref 8 is cited many times in the article. Please provide several more sentence to describe it briefly
Method:
Line 104. Briefly describe how the CPR are analyzed. Is that a high-sensitivity C-reactive protein or a standard CRP test?
Line 132. I’m a little confused about the averaging duration of the ambient pollutant level. The results compared the 1-month period concentration prior to EBC test for SO2, NOX, NO2, NO, 165 and O3, but 3-month average for PM. Why not averaging everything into 3-month period since it’s not significant different for PM 1-month data before and during the pandemic.
Line 139. For a more convincing results, Spearman rank and P-value is often reported with Pearson rank
Results
Line 159. “Subsequently all the results were adjusted to age” Please explain the adjustment process and method.
Table 1. It would be better to also show the correlations/co-variance between each other for the factors in Table 1. I also suggested to investigate the relationship between age/gender/smoker and the three core terms in this study (UFP in EBC, UFP in serum and CRP in serum) within the control group or study group
Reviewer 3 Report
Dear Authors,
the paper has a very good structure and is of interest to the scientific community. However, I would like to raise a relevant issue: I found it rather difficult to understand what is the aim of the study. Keeping in mind the 'correlation is not causation' rule, you study the difference in EBC levels collected on two samples (pre-and-during pandemic) to evaluate the impact of lockdown on the health status (at least respiratory) of people in the region of interest. So is this a long-term exposure study (effect of traffic inqunants on the respiratory system) or a short-term effet (COVID effect on health)? Also, more appropriate statistical tools (e.g., causal inference approaches, regression with both control and policy (e.g., dummies or intervention variables) terms, event studies (more for time series, which are not your case) would be needed for a policy/event impact assessment study. The tools you used may be fine for a descriptive study of pollution effects, but not for a causal study.
Please discuss your objectives in depth in the abstract, introduction, and conclusion, highlighting why the tools you have chosen can be considered statistically valid.
Keep the good job!